# Cloning and Functional Analysis of CsROP5 and CsROP10 Genes Involved in Cucumber Resistance to *Corynespora cassiicola*

**DOI:** 10.3390/biology13050308

**Published:** 2024-04-28

**Authors:** Guangchao Yu, Lian Jia, Ning Yu, Miao Feng, Yue Qu

**Affiliations:** 1College of Chemistry and Life Sciences, Anshan Normal University, Anshan 114007, China; jl_58@163.com (L.J.); yuning361@163.com (N.Y.); fengmiao8826@163.com (M.F.); quyue199209@163.com (Y.Q.); 2Liaoning Key Laboratory of Development and Utilization for Natural Products Active Molecules, Anshan Normal University, Anshan 114007, China

**Keywords:** cucumber, *Corynespora cassiicola*, *CsROP5* gene, *CsROP10* gene, plant defense

## Abstract

**Simple Summary:**

The *CsROP* gene plays a critical role in the regulation of defense responses. We cloned the cucumber genes *CsROP5* and *CsROP10* and identified their structure domains containing two Rho-related guanosine triphosphatases. The *CsROP5* and *CsROP10* genes negatively contributed to defense resistance to *Corynespora cassiicola* by regulating the ROS signaling pathway, the ABA signaling pathway, and the *PR* gene.

**Abstract:**

The cloning of resistance-related genes *CsROP5*/*CsROP10* and the analysis of their mechanism of action provide a theoretical basis for the development of molecular breeding of disease-resistant cucumbers. The structure domains of two Rho-related guanosine triphosphatases from plant (ROP) genes were systematically analyzed using the bioinformatics method in cucumber plants, and the genes *CsROP5* (Cucsa.322750) and *CsROP10* (Cucsa.197080) were cloned. The functions of the two genes were analyzed using reverse-transcription quantitative PCR (RT-qPCR), virus-induced gene silencing (VIGS), transient overexpression, cucumber genetic transformation, and histochemical staining technology. The conserved elements of the CsROP5/CsROP10 proteins include five sequence motifs (G1-G5), a recognition site for serine/threonine kinases, and a hypervariable region (HVR). The knockdown of *CsROP10* through VIGS affected the transcript levels of ABA-signaling-pathway-related genes (*CsPYL*, *CsPP2Cs*, *CsSnRK2s*, and *CsABI5*), ROS-signaling-pathway-related genes (*CsRBOHD* and *CsRBOHF*), and defense-related genes (*CsPR2* and *CsPR3*), thereby improving cucumber resistance to *Corynespora cassiicola*. Meanwhile, inhibiting the expression of *CsROP5* regulated the expression levels of ROS-signaling-pathway-related genes (*CsRBOHD* and *CsRBOHF*) and defense-related genes (*CsPR2* and *CsPR3*), thereby enhancing the resistance of cucumber to *C. cassiicola*. Overall, *CsROP5* and *CsROP10* may participate in cucumber resistance to *C. cassiicola* through the ROS and ABA signaling pathways.

## 1. Introduction

*Corynespora cassiicola* is a pathogenic fungus with a wide host range. It causes cucumber target leaf spot (TLS), which is a significant leaf disease in cucumber production [1]. Cassiicolin is the main pathogenic factor of the fungus, which is a small, secreted glycoprotein consisting of six different subtypes: *Cas1*, *Cas2*, *Cas3*, *Cas4*, *Cas5*, and *Cas6* [2,3,4,5,6]. Currently, there is an increasing amount of research on the molecular mechanisms of the interaction between cucumber and *C. cassiicola*. Cucumbers with *C. cassiicola* resistance include the caffeoyl shikimate esterase gene (*CsCSE5*) [7]. *CsSnRK1* overexpression improves cucumber resistance to *C. cassiicola*, and *CsSnRK1* silencing reduces the resistance of cucumber to *C. cassiicola* [8]. Two cucumber genes, *CsMLO1* and *CsMLO2*, negatively regulate its resistance to *C. cassiicola* [9]. Therefore, we speculate that complex networks form during pathogen infection in plants, and different signaling pathways are related to the transcription of some genes in their mediated disease-resistance responses. Research has confirmed that plant Rho-type (ROP) small G proteins are involved in various signal transduction processes, including regulating pollen tube growth and root hair growth and responding to biotic and abiotic stresses [10,11,12,13,14,15]. However, the molecular mechanism of *CsROP* in regulating cucumber resistance to cucumber TLS remains unclear.

Small G protein is a GTP-binding protein that contains an active guanosine triphosphatase (GTPase). According to their functional differences, small G proteins can be categorized into five families: Ras, Rho, Rab, Ran, and Arf. ROP (Rho-related GTPase from plants) belongs to the small G protein Rho subfamily and is a very conserved signaling molecule [16,17]. Yang and other scholars proposed a nomenclature system in 2002 to unify the ROP GTPase in *Arabidopsis* as ROPs [18]. The highly conserved ROP family members have four GTP/GDP-binding domains and one domain that binds to downstream effector proteins [19]. ROP plays a molecular switch role in plant signal transduction, similar to other small G proteins. The GDP-bound inactive form is mediated by guanine nucleotide exchange factor (GEF) to transform into the GTP-bound active form [20]. Small G proteins in the GTP-binding state have only weak GTPase hydrolysis activity, which requires a GTPase-activated protein (GAP), which can efficiently inactivate ROP in the GTP state and convert it into GDP, releasing inorganic phosphorus. Guanine nucleotide dissociation inhibitor (GDI) in the cytoplasm can bind to the C-terminal lipid modification of ROPs, thereby affecting their membrane localization [20]. The activation of ROP is mediated by upstream signals, and ROP also activates downstream targets. The ROP molecular switch can activate numerous downstream cellular activities, including the dynamic assembly of the microfilament cytoskeleton, calcium ion fluctuations, ROS production, and vesicle transport [21,22,23,24]. Research has found that *Arabidopsis AtROP10* is a negative regulator of multiple physiological responses in ABA signaling, including stomatal closure, seed dormancy and germination, the inhibition of root growth, and gene expression [25,26].

The function of ROP is not only restricted to plant physiological process regulation but is also important for plant responses to pathogen invasion. The overexpression of constitutive-activated *GhROP6* can increase the contents of jasmonic acid–isoleucine and lignin in transgenic *arabidopsis* to enhance resistance to *Verticillium wilt* [27]. The activation of *OsRac1* affects the expression of related defense genes to enhance rice resistance to rice blast and bacterial blight [15], and *OsRac1* and the transcription activator RAI1 jointly participate in resistance to rice blast disease [28]. There are also reports that *OsRac4*, *OsRac5*, and *OsRac6* negatively regulate the PTI defense response to rice [29,30]. Barley ROP protein (RACB) can block the invasion of *Blumeria graminis* into plant tissues [31]. In addition, *ROP* can also regulate the establishment of the ETI resistance mechanism. The overexpression of *DNOsRacl* in tobacco can significantly inhibit the production of ROS and HR, which are induced by the resistance gene *Pto* [32]. *StRac1* positively regulates potato resistance to *Phytophthora infestans* by mediating the H_2_O_2_ content [33]. *ROP*-mediated H_2_O_2_ production can be inhibited by DPI (NADPH oxidase inhibitor), indicating that NADPH oxidases, also called respiratory burst oxidase homologs (RBOHs), are located downstream of small G proteins and regulate the production of H_2_O_2_ [34,35]. In plant–pathogen interactions, *AtRBOHD* and *AtRBOHF* are associated with apoplastic ROS generation to regulate their resistance [36,37]. *CAOsRac1* can enhance pathogen-associated molecular pattern (PAMP)-induced ROS accumulation and resistance to pathogens in rice, and *OsRac1* can both induce the accumulation of ROS and inhibit the activity of ROS scavenging enzymes [15,38]. Similarly, the overexpression of the activating *CAHvRac1* in barley can resist infection of powdery mildew by accumulating ROS [39].

Hormone signal transduction is also one of the main pathways of small G protein resistance. The negative regulatory effect of ROPs on ABA response is concluded through the study of plant guard cells. In the absence of exogenous ABA, the overexpression of CA *ROP6* can inhibit ABA-induced stomatal closure in wild-type *arabidopsis* plants. However, the expression of DN-*ROP6* can cause stomatal closure in the leaves of wild-type and ABA mutant *abi-1* [40,41]. *ROP9* and *ROP10* have also been shown to be negative regulators of the ABA response [42]. Treatment with ABA leads to the inactivation of *AtROP3*, the *AtROP10* promoter in *Arabidopsis* [40,43]. *PYLs*, *PP2Cs*, and *SnRK2s* are the three core components of the ABA signaling pathway [44]. *ABI5* is an important leucine zipper (bZIP) transcription factor in the ABA-mediated signal transduction pathway [45]. However, some studies have shown that small G proteins play a negative regulatory role in disease resistance. Zhang et al. (2014) found that heterologous overexpression of the inactivated DN-*ROP1* of *Arabidopsis* in potatoes can significantly enhance their resistance to *Phytophthora infestans*, which is associated with the NADPH oxidase-mediated accumulation of H_2_O_2_ [46]. However, the molecular mechanism of gene resistance to *C. cassiicola* in the signaling pathway of cucumber *CsROPs* has not been explored.

In this study, two *CsROPs* were cloned from cucumber leaves. A sequence analysis of the *CsROPs* revealed that one structure was similar to that of *Arabidopsis ROP5*, named *CsROP5* (Cucsa.322750), and the other structure was similar to that of *Arabidopsis ROP10*, named *CsROP10* (Cucsa.197080). The transient *CsROP5*/*CsROP10* overexpression reduced the host resistance response to *C. cassiicola*, whereas transient *CsROP5*/*CsROP10* silencing enhanced the resistance of cucumber to *C. cassiicola*. Furthermore, changes in the transcript levels of pathogenesis-related proteins, ABA-signaling-pathway-associated genes, or ROS-associated genes mediated via *CsROP10* silencing enhance cucumber resistance to *C. cassiicola*. *CsROP5* silencing regulated the expression of ROS-signaling-related genes and pathogenesis-related proteins, enhancing the defense response to *C. cassiicola*. Therefore, we speculated that *CsROP5* and *CsROP10* may be negative modulators and indirectly involved in cucumber defense responses to *C. cassiicola* through multiple signaling networks.

## 2. Materials and Methods

### 2.1. Plant Materials

The cucumber cultivar *C. sativus* L. cv. Xintaimici was used in this experiment. Xintaimici is susceptible to *C. cassiicola*. All cucumber plants were grown in a nutrient compound of peatsoil/vermiculite (1/2, *v*/*v*) at 25 °C, in a 16 h/8 h light/dark cycle.

### 2.2. Pathogens and Inoculation

*C. cassiicola* of cucumber was streaked on PDA medium (10 g·L^−1^ D-glucose, 10 g·L^−1^ agar, and 200 g·L^−1^ potatoes). Solid clumps of *C. cassiicola* were inoculated on transgenic cucumber cotyledons in a greenhouse. The optimal cultivation temperature for *C. cassiicola* is 25 °C.

### 2.3. Sequence Analysis

The candidate cDNA sequences of *CsROP5* (Cucsa.322750) and *CsROP10* (Cucsa.197080) from susceptible cucumber leaves were cloned. The primers listed in Appendix A were designed for the cDNA sequence of the DNAMAN. The *CsROP5* and *CsROP10* sequences were obtained from the BLAST (http://cucurbitgenomics.org/BLAST, accessed on 5 June 2022) results for similarity analysis (Appendix A). The conserved domains of CsROP5 and CsROP10 were compared with *Arabidopsis thaliana* ROP family members and human ROP family members (Figure 1).

### 2.4. Quantitative RT-qPCR

The expression patterns of candidate genes were analyzed with RT-qPCR using a Rocha instrument. The internal reference gene we selected was the cucumber actin gene [47]. Appendix A shows the specific primers.

### 2.5. Virus-Induced Gene Silencing (VIGS)

Using the pTRV (tobacco rattle virus)-based VIGS technique, we constructed a knockdown vector of the *CsROP5* and *CsROP10* genes. The two sequences were *CsROP5* (from nucleotide 378 to 591 in the *CsROP5* cDNA sequence) and *CsROP10* (from nucleotide 429 to 639 in the *CsROP10* cDNA sequence), which were subcloned in a sense orientation into the pTRV2 vector. The gene primers are shown in Appendix A. 

*Agrobacterium tumefaciens* strain EHA105 was placed in YEP medium with 100 mg·L^−1^ of rifampicin and 50 mg·L^−1^ of kanamycin and grown overnight at 28 °C. Then, 200 μL of the solution cultured overnight was absorbed and transferred to 20 mL of YEP medium with 100 mg·L^−1^ of rifampicin and 50 mg·L^−1^ of kanamycin. The above solution was incubated with shock until reaching OD_600_ = 0.8–1.0. 

*A. tumefaciens* strain EHA105, harboring recombinant vectors (pTRV2, pTRV-*CsROP5*/pTRV-*CsROP10*), was blended with *A. tumefaciens* strain pTRV1 in equal volumes. They were cultured in an induction medium (10 mM MES ethanesulfonic acid, pH 5.7, 10 mM MgCl_2_, and 200 μM acetosyringone) and diluted to OD_600_ = 0.4. Cucumber cotyledons were injected with an impregnation solution separately, including pTRV::00, pTRV::*CsROP5*, and pTRV::*CsROP10* [48].

### 2.6. Construction of the Overexpressing Vector

The *CsROP5* and *CsROP10* cDNA sequences with the stop codon removed were amplified, and then PCR products were inserted between the *SalI* and *BamHI* cleavage sites of the pRI101-GFP vector. The expression of two target genes was induced in cucumber cotyledons under the control of the CaMV 35S promoter. The gene primers are shown in Appendix A. *Agrobacterium tumefaciens* strain EHA105 was placed in YEP medium with shock until reaching OD_600_ = 0.8–1.0. The detailed steps are the same as described above.

The fusion genes of GFP::00, GFP::*CsROP5*, and GFP::*CsROP10* were introduced to *A. tumefaciens* strain EHA 105 and were supplemented with medium of 10 mM MES, 10 mM Mgcl_2_, and 200 μM AS to generate transgenic cucumber cotyledons [48]. The concentration that infiltrates cucumber cotyledons is OD_600_ = 0.4 [48].

### 2.7. Histochemical Analysis

The H_2_O_2_ and O_2_·^−^ levels of the cucumber cotyledons were assessed using 3,3′-diaminobenzidine (DAB) and nitrotetrazolium blue chloride (NBT) [49]. The H_2_O_2_ test was performed by soaking the cucumber cotyledon in 1 mg·mL^−1^ DAB for 8 h. The O_2_^−^ test was performed by soaking the cucumber cotyledon in 0.1% NBT for 5 h. Then, ethanol/lactic acid/glycerol (3:1:1) was added to the cucumber cotyledons, and they were boiled for 20 min, transferred to 95% ethanol, and stored at 4 °C.

## 3. Results

### 3.1. Cloning and Sequence Analysis of CsROP5 and CsROP10

Two full-length cDNA sequences of *CsROP5* and *CsROP10* were cloned from cucumber leaves. The analysis showed that the *CsROP5* cDNA encoded a 21.56 kDa protein with 197 amino acid residues and the *CsROP10* cDNA encoded a 23.42 kDa protein with 210 amino acid residues (Appendix A). An analysis of the protein sequence showed that CsROP5 and CsROP10 contain ROP-conserved domains (Figure 1). The CsROP5 and CsROP10 proteins were quite similar to the *Arabidopsis thaliana* ROP family members, which contained the highest conservation rate, including five sequence motifs (G1–G5) that were composed of a central six-stranded β-sheet (β1–β6) surrounded by five α-helices (α1–α5), the helix αi of the Rho insert, and a short helical structure (η1) upstream of the insert (Figure 1). Additionally, the sequences of CsROP5 and CsROP10 were clearly distinct from Rac1 as well as RhoA. The sequences of CsROP5 and CsROP10 were significantly distinct from the Rac1 and RhoA sequences in that there was a possible recognition site (SYR) for serine/threonine kinases, which was produced by the conserved arginine present in the downstream ROP protein of switch II [19].

### 3.2. CsROP5 and CsROP10 Silencing in Cucumber Enhances Its Resistance to C. cassiicola

In order to investigate whether the transcript levels of *CsROP5* and *CsROP10* affect the resistance of cucumber to *C. cassiicola*, the VIGS technique was used to knockdown the *CsROP5*/*CsROP10* transcript in the susceptible cultivar Xintaimici (Figure 2). The 3′-terminal fragment specific to *CsROP5* and *CsROP10* was inserted into the restriction site of pTRV2, which generated the recombinant structures of pTRV2*-CsROP5* and pTRV2-*CsROP10* (Figure 2A). After 7 days of inoculation with pTRV2-*CsROP5*, pTRV2-*CsROP10*, and TRV:00 (as a control), the cucumber cotyledons of the infected plants showed chlorotic mosaic symptoms (Figure 2B), proving that these inoculated plants were successfully infected with TRV. Meanwhile, the RT-qPCR confirmed the effective silencing of *CsROP5* and *CsROP10* (Figure 2C). These results indicate that *CsROP5* and *CsROP10* were almost completely unexpressed in cucumber plants after silencing.

Subsequently, *C. cassiicola* was inoculated on cucumber cotyledons to evaluate the defense resistance of *CsROP5* and *CsROP10* (Figure 3). To investigate the role of *CsROP5* and *CsROP10* in the defense against *C. cassiicola*, we tested the transgenic cucumber cotyledons after 5 days of pathogen infection. The results of the lesion size measurement showed that the cucumber cotyledons were comparable between the non-injected cucumber cotyledons and the cucumber cotyledons inoculated with TRV:00. However, the lesion areas on the cotyledons of *CsROP5*-silencing and *CsROP10*-silencing plants were 0.40–0.45 cm, while the average lesion areas of the non-injected cucumber cotyledons and the TRV:00-injected groups were 0.60–0.63 cm. These results show that *CsROP5* and *CsROP10* silencing in Xintaimici plants enhanced their defense response to *C. cassiicola*.

### 3.3. CsROP5 and CsROP10 Overexpression in Cucumber Cotyledons Impairs Resistance to C. cassiicola

For further evaluation of the roles of *CsROP5* and *CsROP10* in the defense responses, two overexpression vectors containing pRI101:GFP-*CsROP5* and pRI101:GFP-*CsROP10* were generated. The constructed recombinant plasmids (GFP::*CsROP5*/GFP::*CsROP10*) and empty plasmids (GFP::00) were transformed into *Agrobacterium tumefaciens* to be transferred into cucumber cotyledons (Figure 4). The PCR analysis using chimeric primers between the *CsROP5/CsROP10* and GFP fusion genes showed that the introduced transgene could be detected in the cucumber cotyledons (Figure 4A; Figure 4B). Meanwhile, the RT-qPCR results show that the transcript abundances of *CsROP5/CsROP10* were significantly increased compared to the GFP::00-injected plants (Figure 4C). Our results demonstrate that *CsROP5* and *CsROP10* were successfully transiently overexpressed in cucumber cotyledons.

Following inoculation with *C. cassiicola* for 5 d, comparing the non-injected and GFP::00-injected plants, the defense resistance levels to *C. cassiicola* of *CsROP5/CsROP10*-overexpressing cucumber cotyledons had significantly reduced (Figure 5). The lesion size measurements in the cotyledons of *CsROP5*-overexpressing and *CsROP10*-overexpressing plants infected with *C. cassiicola* ranged from 0.79 to 0.81 cm, while the average lesion areas of the non-injected cucumber cotyledons and the GFP::00-injected groups were 0.60–0.62 cm. The above results show that transient overexpression of *CsROP5* and *CsROP10* reduced cucumber resistance to *C. cassiicola*. These results indicate that *CsROP5* and *CsROP10* negatively regulate cucumber resistance response to *C. cassiicola*.

### 3.4. CsROP5 and CsROP10 Modulate the Expression Levels of Defense-Related Genes after C. cassiicola Challenge in Transgenic Plants

To investigate whether *CsROP5*/*CsROP10* expression influenced defense-related genes in cucumber response to *C. cassiicola*, the expression patterns of two pathogenesis-related genes were analyzed in transgenic cucumbers (Figure 6). The results show that the expression of *CsPR2* was significantly increased in *CsROP5*/*CsROP10*-silencing cotyledons compared to TRV::00-infected plants, and the transcript level of *CsPR2* was only decreased in *CsROP5*-overexpressing cucumber cotyledons after *C. cassiicola* inoculation for 5 days. Additionally, the transcript levels of *CsPR3* were also significantly induced in *CsROP5*/*CsROP10*-silencing cotyledons, and the transcript levels of *CsPR3* were significantly declined in *CsROP5*/*CsROP10*-overexpressing cucumber cotyledons. In summary, we speculate that *CsRP* might participate in the defense reaction through the *CsROP5*/*CsROP10*-induced defense pathway.

### 3.5. ROS Homoeostasis Is Crucial for CsROP5- and CsROP10-Mediated Resistance against C. cassiicola

To investigate whether *CsROP5* and *CsROP10* mediate the homeostasis between ROS scavenging and production to affect cucumber resistance to *C. cassiicola*, we conducted the following experiments (Figure 7). H_2_O_2_ and O_2_^−^ were detected using the DAB and NBT staining methods in transgenic plants (Figure 7B). After 12, 24, and 48 h of inoculation with *C. cassiicola*, the accumulation of H_2_O_2_ and O_2_^−^ in *CsROP5*/*CsROP10*-silencing plants was increased compared to that in TRV::00 plants, with more brown and blue spots on the leaves. However, little H_2_O_2_ and O_2_^−^ was detected at 12 h with *C. cassiicola* inoculation in the *CsROP5*/*CsROP10*-overexpressing plants and the GFP::00 plants. After 24 and 48 h, the accumulation of H_2_O_2_ and O_2_^−^ was lower in the *CsROP5*/*CsROP10*-overexpressing cucumber than that in the GFP::00 cucumber. Overall, the accumulation of H_2_O_2_ and O_2_^−^ was significantly higher in the silencing cucumber than that in the overexpressing cucumber after *C. cassiicola* infection. Following *C. cassiicola* inoculation for 5 days, two ROS-formation-related genes, *CsRbohD* (Cucsa.340760) and *CsRbohF* (Cucsa.107010), were closely linked to the transcript levels of *CsROP5* and *CsROP10* in the cucumber cotyledons (Figure 7A). The transcript levels of the above two genes were significantly higher in the *CsROP10*-silencing plants than in the TRV::00 plants and were lower in the *CsROP10*-overexpressing plants. However, the transcript levels of *CsRbohD* and *CsRbohF* were only elevated in the *CsROP5*-silencing plants compared to the controls, but nothing changed in the *CsROP5*-overexpressing plants. These results imply that ROS signaling was triggered in the *CsROP10*-silencing plants, mediating their resistance to *C. cassiicola*.

### 3.6. CsROP5 and CsROP10 Modulate Abscisic Acid (ABA)-Signaling Components

ROP GTPases are important signaling proteins that regulate ABA-related responses [21,40,43]. Therefore, the transcript levels of genes related to the ABA signaling pathway were detected in *CsROP5*/*CsROP10*-silencing plants and *CsROP5*/*CsROP10*-overexpressing plants after *C. cassiicola* inoculation for 5 days (Figure 8). The RT-qPCR analysis showed that the transcript levels of *CsPYL2* (JF789829), *CsSnRK2.2* (JN566071), and *CsABI5* (XM_004149176.2) were increased in the *CsROP10*-silencing plants but decreased in the *CsROP10*-overexpressing plants. However, the transcript level of *CsPP2C* (JN566067) showed the opposite trend, resulting in a significantly suppressed transcript level of *CsPP2C* in the *CsROP10*-silencing plants and a more markedly induced level in the *CsROP10*-overexpressing plants than in the control plants. In addition, we also conducted transcript-level detection of genes related to the ABA pathway in *CsROP5* transient transgenic plants. Compared to the control levels, the transcript level of *CsPP2C* was decreased in the *CsROP5*-silencing plants and increased in the *CsROP5*-overexpressing plants. The transcript level of *CsPYL2* was only improved in the *CsROP5*-overexpressing plants. However, there was almost no change in the transcript levels of *CsABI5* and *CsSnRK2.2* in the *CsROP5* transient transgenic plants. These results suggest that *CsROP10* negatively regulated the transcript levels of *CsPYL2*, *CsSnRK2.2*, and *CsABI5* and positively regulated the transcript level of *CsPP2C*. Together, these results suggest that *CsROP10* was a negative regulator and improved the transcript level of ABA-signaling component genes to enhance the cucumber defense response to *C. cassiicola*. However, the expression regulation of ABA signaling pathway genes was not directly related to the transcript level of *CsROP5*, indicating that there might be other defense mechanisms of *CsROP5* in the cucumber—*C. cassiicola* interaction.

## 4. Discussion

The CDS regions of two *CsROP* genes, namely *CsROP5* and *CsROP10*, were identified and cloned in cucumber cotyledons. A sequence analysis showed that CsROP5 and CsROP10 were distinct from other RHO GTPases in several aspects. First, the highly conserved effector domain (domain II) of CsROP5 and CsROP10 contained the highly conserved effector domain (domain I and domain II). Second, there were RHO insertion regions (structural domains V) in CsROP5 and CsROP10, which interacted with the RHO effectors [50]. In addition, the sequence analysis showed that CsROP5 and CsROP10 contained two putative serine/threonine phosphorylation sites, SYR and SSK. The above domain modules were consistent with the ROPs of *Arabidopsis* [17]. It has been confirmed that Arabidopsis, rice, maize, and other species contain multiple ROP proteins [20,51,52]. The ROP family also has diversity, with the highly variable region (HVR) defined by the region with the most variable C-terminus of the ROP protein [53,54,55]. In our study, an HVR similar to that of ROP family genes was predicted in cucumber plants. Meanwhile, a functional analysis found that barley Rac/ROP G-protein family members were susceptible to the powdery mildew fungus [55]. The study demonstrated that *CsROP5* and *CsROP10* genes might mediate pathogen resistance in cucumber.

Increasing evidence shows that ROPs are an indispensable regulatory factor in plant immune signaling [56]. *OsRac4* and *OsRac5* are negative regulators of blast resistance [30]. Three members of the Rac/ROP family in barley—*HvRACB*, *HvRAC3*, and *HvROP6*—play a negative regulatory role in powdery mildew fungus [55,57]. Currently, the role of the *ROP* gene is unclear in cucumber–*C. cassiicola* interactions. Due to constriction in cucumber transgenic technology, research on the molecular mechanism of *ROP* gene resistance to disease has been hindered in cucumbers. At present, the experimental method of transient *agrobacterium* infection of cucumber cotyledons has matured [48]. This study on *CsROP5* silencing and *CsROP10* overexpression was further carried out using an experimental method of *agrobacterium* infiltration into cucumber cotyledons. The infection assays insinuated that *CsROP5* and *CsROP10* played an important role in the resistance of cucumber to *C. cassiicola*. *CsROP5*/*CsROP10*-silencing plants exhibited strongly enhanced resistance to *C. cassiicola*, while *CsROP5*/*CsROP10*-overexpressing plants showed significantly reduced resistance to *C. cassiicola*. In addition, ROS accumulation is required for the plant defense mechanism when part of a plant is subjected to *C. cassiicola* attack [58]. In our research, *ROP* might have triggered the outbreak of ROS, thereby enhancing the resistance to *C. cassiicola* infection in cucumber cotyledons. ROS accumulation also occurred earlier in the *CsROP5*/*CsROP10*-silencing plants than in the *CsROP5*/*CsROP10*-overexpressing plants after *C. cassiicola* infection. Thus, we preliminarily speculate that the responses of the plants to *C. cassiicola* stress may have revealed *CsROP5* and *CsROP10* as negative regulatory factors.

As is well known, the most influential enzyme for ROS production is NADPH oxidase. The interaction between *OsRac1* and the N-terminal domain of *OsrbohB* promotes the accumulation of ROS in rice [59,60]. The interaction between *DN*-*AtROP1* and *StrbohD* (a potato NADPH oxidase) induces H_2_O_2_ accumulation to negatively regulate resistance to potato *Phytophthora infestans* [46]. In similar cases, ROS-producing NADPH oxidases are affected by *AtrbohD* and *AtrbohF* genes in response to pathogen infection [36,37,61]. The overexpression of *CA OsRac1* can correspondingly increase the level of ROS and cause a response similar to HR, thereby increasing the resistance of rice to *Magnaporthe oryzae* and *Blumeria graminis* [39]. Furthermore, we found that the expressions of *CsRbohD* and *CsRbohF* were increased in *CsROP5*/*CsROP10*-silencing cucumbers, whereas the expressions of *CsRbohD* and *CsRbohF* were suppressed in *CsROP10*-overexpressing cucumbers. Compared to the control, the accumulation of H_2_O_2_ and O_2_^−^ was reduced in *CsROP10*-silencing cucumbers after *C. cassiicola* inoculation, and similar results were observed in *CsROP5*-silencing cucumbers. Therefore, it is highly likely that small G proteins have similar ways of generating ROS in cucumbers, and the differences in the binding affinity or activation ability of *CsROP5*/*CsROP10* to the cucumber RBOHB homologs led to differences in ROS production. Additionally, these results suggest that ROS signaling might be pre-activated in *CsROP5*/*CsROP10*-silencing cucumbers to enhance defense resistance against *C. cassiicola*.

In addition, pathogenesis-related proteins (PRs), including β-1,3-glucanase (*PR2*) and chitinase (*PR3*) [62,63], are a type of protein produced in plants under pathogen infection to regulate defense responses. Previous research has demonstrated that the expression of *arabidopsis* pathogenesis-related proteins improve resistance to *B. cinerea* interactions [64]. The Rac/ROP plays a key role in defense systems, such as regulating the expression of pathogen-related genes and lignin biosynthesis [65,66,67,68]. In our study, the expression patterns of defense-related *CsPR2* and *CsPR3* were altered in *CsROP5*/*CsROP10*-induced disease resistance. The treatment of *CsROP5*/*CsROP10*-silencing plants primed defense responses against *C. cassiicola* in cucumber plants by inducing higher expression of the defense-related genes *CsPR2* and *CsPR3*. This finding indicates that other resistance mechanisms exist in response to *C. cassiicola*; *CsROP5*/*CsROP10* primarily mediated the defense resistance via the PR proteins. In summary, *CsROP5*-silencing and *CsROP10*-silencing regulated ROS signaling pathways may jointly activate downstream pathogenesis-related gene expression against *C. cassiicola*.

ABA is a key hormone that regulates plant responses to many types of adaptive stress [69]. ABA can actively regulate stomatal closure against pathogens or induce callose deposition when pathogens evade the first line of defense, thereby regulating plant defense in the early stages of infection [70,71]. In addition, research on plant guard cells has shown that ROPs can negatively regulate ABA signaling [40]. Research has found that *ROP9* and *ROP10* are negative regulators of ABA reactions. Treatment with ABA resulted in the inactivation of *AtROP3* (*AtRac1*) and *AtROP10* promoters in *Arabidopsis* [40,43]. It was found that *CsPYL2*, *CsPP2C2*, and *CsSnKR2.2* are involved in transducing the ABA signal in regulating plant development and drought stress [44]. A comprehensive expression analysis of ABA signal transduction genes in cucumber revealed that *CsROP10* inhibited the expressions of *CsPYL2* and *CsSnKR2.2* and facilitated the expression of *CsPP2C2* in response to *C. cassiicola.* These results are consistent with the results of Ulferts, who reported that ABA negatively interfered with the basal defense of barley against *Magnaporthe oryzae* [72]. In addition, compared to the control, the transcript level of *CsABI5* was upregulated by seven-fold at 5 dpt under *C. cassiicola* in the *CsROP10*-silencing plant, although the transcript level of *CsABI5* was significantly downregulated in the *CsROP10*-silencing plant (Figure 8). Similar to these findings, *CsABI5* genes were highly induced in *CsMLO1*-silencing cucumber cotyledons after *C. cassiicola* infestation [9]. The above results also coincide with the interaction effect in *CsROP10* and *C. cassiicola* via mediating the ABA signaling pathway. ABA binds to receptors (RCAR/PPYR/PYL), leading to phosphorylating *PP2C* and activating protein kinase. Additionally, more and more evidence suggests that ROP and ABA form a negative feedback loop, where ABA signaling inhibits ROP activation and ROP signaling inhibits ABA response; for example, *ROP10* and *ROP11* inhibit ABA signal transduction through physical interactions with ABA negative regulatory factors, ABA insensitivity 1 (*ABI1*), and *ABI2 PP2C* phosphatase [73,74]. Therefore, we speculate that *CsROP10* also may interact with ABI proteins to inhibit ABA signaling pathways in cucumber plants. The elevated kinase activates NADPH oxidase and elevates ROS production [75]. Furthermore, RBOHC is activated by activated ROPs, and this change leads to the generation of ROS, which causes changes in the intracellular and extracellular pH and intracellular Ca^2+^ levels [76,77]. Research has found that ROS- and ABA-signaling-related genes are related to plant immunity, thereby improving the resistance of cucumber to *C. cassiicola* in *CsMLO1*-silencing plants [9]. Thus, our results suggest that *CsROP10* participates in pathogen resistance by employing the ABA-mediated ROS signaling pathway and the *R* gene pathway associated with *CsROP10*. However, the change in the transcript level of *CsROP5* had no regular effect on the expression of ABA-related genes. This finding further suggests that the involvement of *CsROP5* in the defense of *C. cassiicola* may be related to ROS signaling and defense-related genes.

## 5. Conclusions

In conclusion, two cucumber genes, *CsROP5* and *CsROP10*, negatively contribute to the defense response to *C. cassiicola. CsROP10* can regulate the expression of ABA-signaling-related genes, thereby promoting the production and accumulation of ROS genes, subsequently modulating the expression of pathogenesis-related proteins and ultimately leading to enhanced resistance to *C. cassiicola*. Cucumber *CsROP5* regulated the expression of ROS-signaling-related genes and pathogenesis-related proteins, enhancing the defense response to *C. cassiicola*. Therefore, this study indicates that the plant defense response is mediated by the interaction of the *CsROP5* and *CsROP10* genes between cucumber and *C. cassiicola*.

## Figures and Tables

**Figure 1 biology-13-00308-f001:**
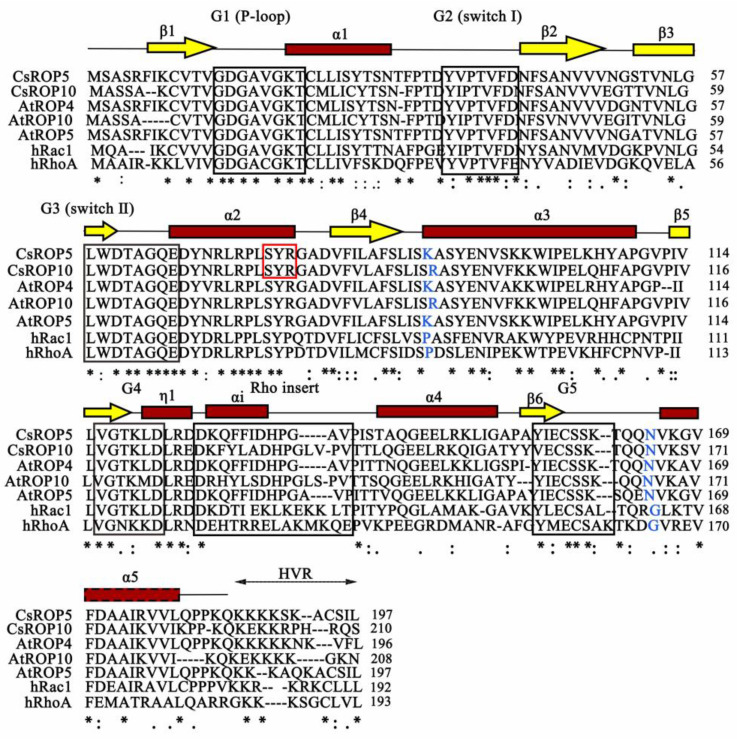
Sequence and structural features of *Arabidopsis thaliana* AtROP4 (Q38937), AtROP5 (BT005217), AtROP10 (BT005228), human hRac1 (NP_008839), and hRhoA (NP_001655). Conserved elements include G1–G5 of the G domain and the hypervariable region (HVR). Red box: putative phosphorylation site; consensus line: * identical residues; : conserved substitutions; . semi-conserved substitutions.

**Figure 2 biology-13-00308-f002:**
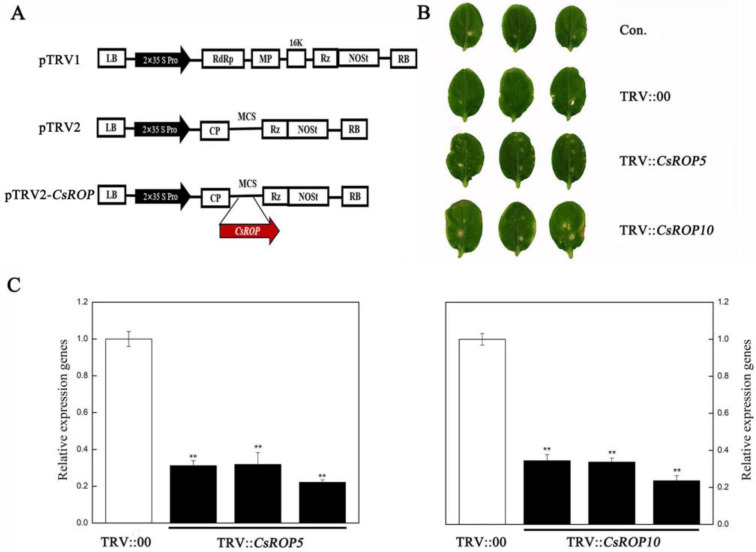
Silencing of *CsROPs* in cucumber plants. (**A**) Construction diagram of *CsROP5* silencing and *CsROP10* silencing. (**B**) Phenotypic analysis in silencing plants. Con. indicates non-injected plants. (**C**) Using RT-qPCR, we identified *CsROP5*-silencing and *CsROP10*-silencing plants. TRV::00: injection of empty vector plant. Asterisks indicate significant difference (Student’s *t*-test, ** *p* < 0.01).

**Figure 3 biology-13-00308-f003:**
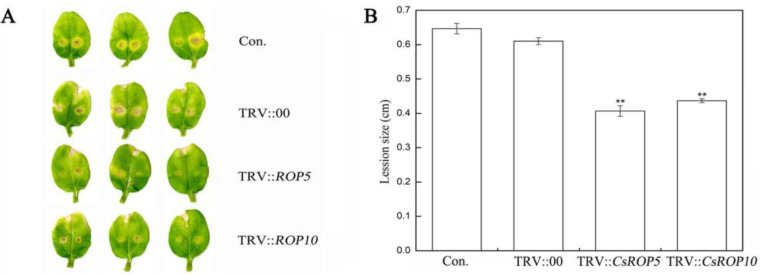
Demonstrated enhanced resistance to *C. cassiicola* in silenced plants. (**A**) The phenotypes of cucumber cotyledons 5 days after inoculation with *C. cassiicola*. (**B**) The lesion size was determined at cucumber cotyledons. Error bars represent standard deviations from three independent replicates. Asterisks indicate significant difference (Student’s *t*-test, ** *p* < 0.01).

**Figure 4 biology-13-00308-f004:**
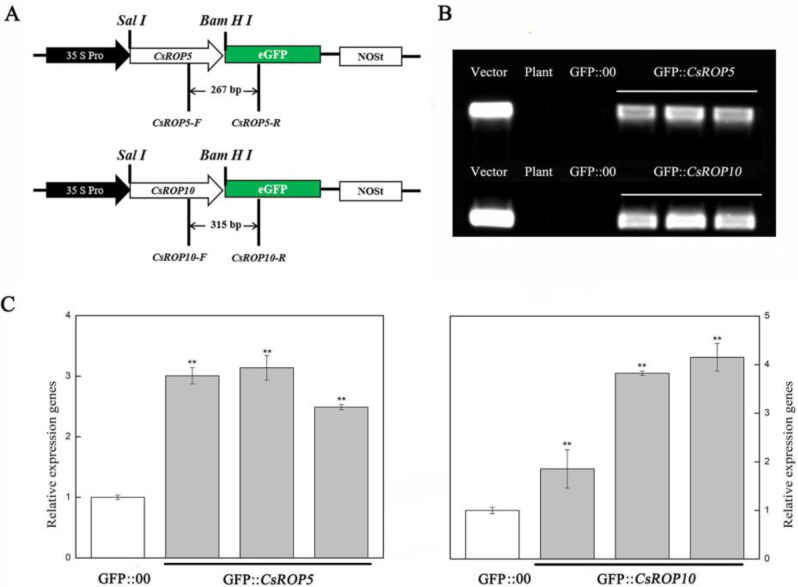
Demonstrated decreased resistance to *C. cassiicola* in overexpressed plants. (**A**) Schematic of the *CsROP5*-GFP and *CsROP10*-GFP constructs. (**B**) Chimeric PCR identification proved that *CsROP5* and *CsROP10* were successfully inserted into the vector. Vector, recombinant plasmid; plant, non-transgenic cucumber; GFP:00, empty vector; GFP::*CsROP5*, *CsROP5* transient overexpressing in cucumbers; GFP::*CsROP10*, *CsROP10* transient overexpressing in cucumbers. (**C**) Using RT-qPCR, we identified *CsROP5*-overexpressing and *CsROP10*-overexpressing plants. GFP::00: injection of empty vector plant. Asterisks indicate significant difference (Student’s *t*-test, ** *p* < 0.01).

**Figure 5 biology-13-00308-f005:**
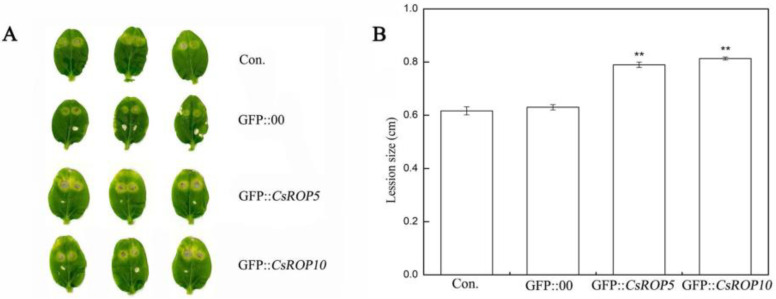
Identification of disease resistance of *CsROP5*-overexpressing and *CsROP10*-overexpressing plants. (**A**) The phenotypes of cucumber cotyledons at 5 days after inoculation with *C. cassiicola*. (**B**) The lesion sizes of cucumber cotyledons at 5 days after inoculation with *C. cassiicola*. Error bars represent standard deviations from three independent replicates. Asterisks indicate significant difference (Student’s *t*-test, ** *p* < 0.01).

**Figure 6 biology-13-00308-f006:**
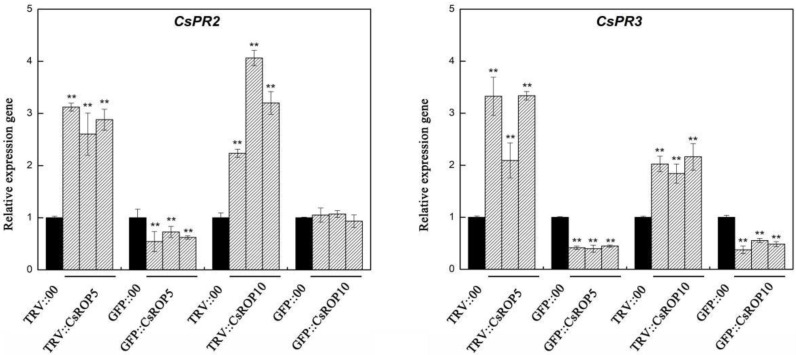
Relative mRNA transcript levels of defense-related genes of *CsROP5* and *CsROP10* transgenic plants as determined by RT-qPCR. Error bars represent standard deviations from three independent replicates. Asterisks indicate significant difference (Student’s *t*-test, ** *p* < 0.01).

**Figure 7 biology-13-00308-f007:**
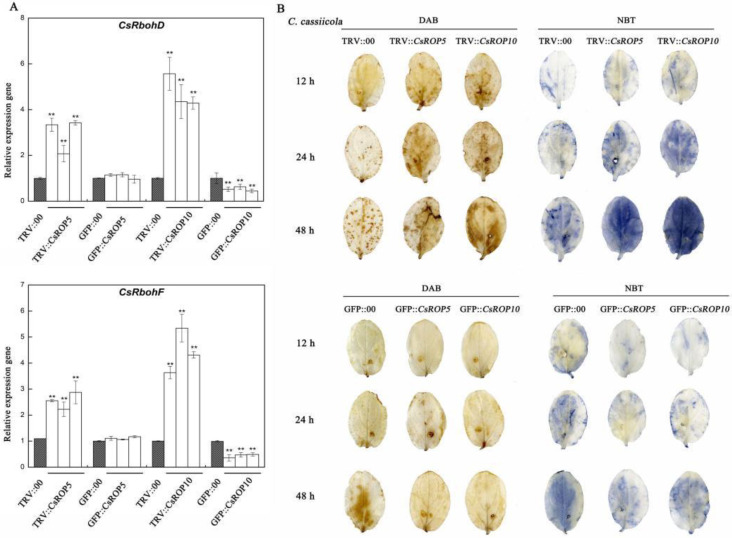
ROS signaling was modulated in transgenic plants after inoculation with *C. cassiicola*. (**A**) Relative mRNA transcript levels of *CsRbohD* and *CsRbohF* in *CsROP5*/*CsROP10* transgenic plants as determined by RT-qPCR. Asterisks indicate significant difference (Student’s *t*-test, ** *p* < 0.01). (**B**) DAB and NBT staining of H_2_O_2_ and O_2_^−^ generation in transgenic plants after *CsROP5* and *CsROP10 Agrobacterium infiltration*. Data are means of three biological replicates of per variety.

**Figure 8 biology-13-00308-f008:**
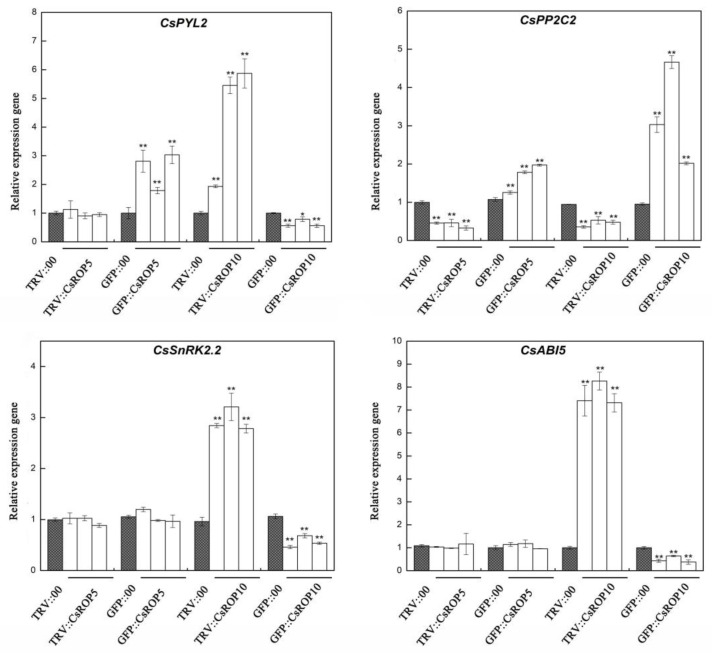
Relative mRNA transcript levels of ABA-related genes in *CsROP5* and *CsROP10* transgenic plants as determined by RT-qPCR. Error bars represent standard deviations from three independent replicates. Asterisks indicate significant difference (Student’s *t*-test, * *p* < 0.05 or ** *p* < 0.01).

## Data Availability

The research data used in the article has been provided in the Appendix A.

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
