# Peer review of "Cloning and Functional Analysis of CsROP5 and CsROP10 Genes Involved in Cucumber Resistance to Corynespora cassiicola"

_biology, 2024, doi:10.3390/biology13050308_

Round 1
Reviewer 1 Report
Comments and Suggestions for Authors
Dear authors,
You have done a good job contributing towards the fight against plant pathogen for a sustainable crop production. I find your study to be well-conducted and potentially impactful for the field of molecular breeding and disease resistance in cucumbers.
With the aim of publication I have some general and specific comments and suggestions for revision.
General comments:
1- The manuscript is well structured, making it easy to follow the sections. However, there are several areas where clarity could be improved. This issues is consistent across the entire manuscript. I recommend rewriting of this manuscript with much attention to the English (sentence structure and syntax, grammar). I have pointed out some of these in my specific comments.
2- It would be helpful if you could provide more detailed protocols for the VIGS and transient overexpression experiments, including information on the vectors and specific procedures used.
3- Your findings have significant implications for cucumber molecular breeding programs aimed at enhancing disease resistance. The identification and functional characterization of CsROP5 and CsROP10 provide valuable insights into the molecular mechanisms underlying cucumber defense responses. However, I suggest you further elaborate on potential future directions in your discussion section. For example, consider discussing potential additional downstream targets of CsROP5 and CsROP10 or investigating their interactions with other signaling pathways to provide a more comprehensive understanding of their roles in cucumber resistance.
Specific comments:
1- Line 1-2: The title reads: CsROP5 and CsROP10 are negative regulator the cucumber defense resistance to Corynespora cassiicola what if "CsROP5 and CsROP10 are negative regulator for cucumber defense resistance to Corynespora cassiicola" OR "Cloning and Functional Analysis of CsROP5 and CsROP10 Genes Involved in Cucumber Resistance to Corynespora cassiicola"
4- I suggest figure S1 and S2 be move to the supplementary folder.
2-Line 13: delete ..."in"...
3- Lines 14 to 16: Sentence needs correction with right tense and sentence structure.
4- Line 18: action mechanism change to "mechanism of action"
5- Line 18 and 19: Consider rewriting "development of cucumber disease resistant molecular breeding".
Thank you for your attention to these comments.
Comments on the Quality of English LanguageI recommend the manuscript for publication following thorough revisions capitalising on the English language, sentence structure, syntax and grammar.
It hard for the reader to understand
Reviewer 2 Report
Comments and Suggestions for Authors
This study shows how genes can act in different ways when overexpressed and silenced against generating resistance in cucumber against C.cassiicola.
It's interesting how these two genes CsROP5 and CsROP10 when knock-downed also affect various primary and secondary defenses in the plant.
Some minor revisions:
Table S1 and S2: Do add Tm and amplicon size for all primers listed
All figures and graphs: Enlarge the figures/graphs and increase the font size as it's hard to read on a printed manuscript
Reviewer 3 Report
Comments and Suggestions for Authors
This manuscript presents two cucumber CsROP5 and CsROP10 genes were negatively contributes to defense response to C. cassiicola. CsROP5 participates in ROS signaling pathways and CsROP10 participates in ABA and ROS signaling pathways. Some evidence is provided in the manuscript, but more evidence needs to be added, and many details need to be revised.
Specific comments:
(1) At present, VIGS technology in many cucurbitaceae crops can realize the observation of true leaves, why in this manuscript, only the observation of cotyledon. The observation of true leaves would be more convincing.
(2) In addition to the measurement of the size of the lesion area, other disease indicators should also be observed.
(3) It is mentioned in the manuscript that genes related to ABA signaling pathway and ROS signaling pathway are related to plant immunity, and VIGS evidence is needed to support them.
(4) CsROP5 and CsROP10 gene cloning why not clone the last three bases.
Details:
(1) Line 49, “networksduring” is supposed to be “networks during”.
(2) Line 65, “into tthe GTP-bound active form” is supposed to be “into the GTP-bound active form”.
(3) Line 152, “knockout” is supposed to be “knockdown”.
Many details require further examination.
Round 2
Reviewer 3 Report
Comments and Suggestions for Authors
The authors revised my questions very well. I have no futher suggestions.